# Reg-1α, a New Substrate of Calpain-2 Depending on Its Glycosylation Status

**DOI:** 10.3390/ijms23158591

**Published:** 2022-08-02

**Authors:** Marie-Christine Lebart, Françoise Trousse, Gilles Valette, Joan Torrent, Morgane Denus, Nadine Mestre-Frances, Anne Marcilhac

**Affiliations:** 1MMDN, Univ Montpellier, EPHE, INSERM, 34095 Montpellier, France; francoise.trousse@umontpellier.fr (F.T.); joan.torrent@inserm.fr (J.T.); morgane.denus@umontpellier.fr (M.D.); nadine.frances@umontpellier.fr (N.M.-F.); anne.marcilhac@umontpellier.fr (A.M.); 2EPHE, PSL Research University, 75014 Paris, France; 3LMP, Univ Montpellier, 34095 Montpellier, France; gilles.valette@umontpellier.fr; 4INM, Univ Montpellier, INSERM, 34095 Montpellier, France

**Keywords:** calpain, Reg-1α, trypsin, cleavage, fibril

## Abstract

Reg-1α/lithostathine, a protein mainly associated with the digestive system, was previously shown to be overexpressed in the pre-clinical stages of Alzheimer’s disease. In vitro, the glycosylated protein was reported to form fibrils at physiological pH following the proteolytic action of trypsin. However, the nature of the protease able to act in the central nervous system is unknown. In the present study, we showed that Reg-1α can be cleaved in vitro by calpain-2, the calcium activated neutral protease, overexpressed in neurodegenerative diseases. Using chemical crosslinking experiments, we found that the two proteins can interact with each other. Identification of the cleavage site using mass spectrometry, between Gln^4^ and Thr^5^, was found in agreement with the in silico prediction of the calpain cleavage site, in a position different from the one reported for trypsin, i.e., Arg^11^-Ile^12^ peptide bond. We showed that the cleavage was impeded by the presence of the neighboring glycosylation of Thr^5^. Moreover, in vitro studies using electron microscopy showed that calpain-cleaved protein does not form fibrils as observed after trypsin cleavage. Collectively, our results show that calpain-2 cleaves Reg-1α in vitro, and that this action is not associated with fibril formation.

## 1. Introduction

Reg-1α [1,2] has been known for many years as PSP (Pancreatic Stone Protein) and lithostathine due to its abundance in pancreatitis and capacity to inhibit spontaneous CaCO_3_ precipitation, respectively [3,4]. The name of Reg-1α (regenerating islet-derived protein 1-alpha) appeared afterwards due to its role in gene activation in regenerating beta pancreatic islets. In addition to its expression in pancreatic cells, the protein was also reported in other peripheric organs, mainly the gastrointestinal tract. The protein, a secreted glycoprotein of 17 kDa, belongs to a large family of proteins which comprises four groups based on their primary structure homology. The particularity of these members is the presence of a C-type lectin domain linked to a short N-terminal peptide. The latter can be generated by trypsin hydrolysis and was shown to be essential to the inhibitory activity of the protein on CaCO_3_ crystal growth [5], or the differentiation of rodent telencephalic neuronal precursors during brain development [6]. Moreover, biochemical studies have revealed that Reg-1α is O-glycosylated on a unique site located in this specific part of the molecule, Thr^5^ [7,8].

Reg-1α is expressed under normal and pathological circumstances, associated with diabetes, cancer, and inflammation. Most interesting is the reported expression of the protein in the brain and cerebrospinal fluid of patients with very early stages of Alzheimer’s disease [9,10]. We recently demonstrated that in the context of neurodegenerative disease Reg-1α increases Tau phosphorylation. Additionally, it promotes Tau deposition and the formation of intracellular aggregates where Reg-1α colocalizes with phosphorylated-Tau [11]. Moreover, the protein was shown in vitro to form fibrils [12], which result from a proteolytic cleavage between Arg^11^-Ile^12^ equivalent to the action of trypsin, with structural characteristics reminiscent of amyloid fibrils [12,13], although distinct [14]. At present, the role of the proteolytic cleavage of Reg-1α is poorly understood and the protease responsible for such cleavage in the central nervous system is currently unknown.

The overexpression of Reg-1α in the brain in neurodegenerative diseases, and its reported limited cleavage associated with calcium homeostasis disturbance, has led us to focus on a particular class of proteases, the calpains. Indeed, these calcium-activated enzymes function by making specific limited cuts in proteins so that they can modulate/modify their substrate activity rather than serving as degradative enzymes. Compared to the proteasome or autophagic system, calpains are unique in that they directly recognize and interact with their substrates. Thus, in this context where trypsin is absent from the brain, calpains, referred to as modulator proteases [15], appear as a strong potential candidate for Reg-1α cleavage. Among the numerous (15 in human) members of the family, two isoforms are ubiquitously expressed, namely calpain-1 and calpain-2 (for review see [16,17]). The distinction between the two isoforms has been difficult, both in substrates selectivity and functions. However, in the last decade the notion has emerged that calpain-1 and -2 play opposite functions in the brain, calpain-1 in synaptic plasticity and neuroprotection, and calpain-2 in neurodegeneration [18,19,20]. The proteases were previously called micro-(calpain1) and milli-(calpain2) calpains due to the concentration needed for their activation in vitro—3–40 µM and 0.4–0.8 mM, respectively, although these concentrations are order of magnitude higher than the resting calcium concentration (multi-nanomolar) inside the cell.

The two calpains’ isoforms are heterodimers, composed of a large isoform specific subunit of 80 kDa encompassing the catalytic triad (identity >70% between the catalytic subunits of the two isoforms) and a common small regulatory subunit of 28 kDa, required for the stability of the molecule. The resolution of the three dimensional structures of calpain-2 in the absence [21,22] and in the presence [23,24] of calcium has provided insight into the understanding of the structure–function relationship and activation of the enzyme. In particular, the catalytic triad of the enzyme, which includes the essential Cys (Cys^105^ in calpain-2), becomes functional upon conformational changes induced by calcium binding.

The list of potential substrates is long, with spectrin and its breakdown products being widely used as markers of calpain activation [25]. The wide range of protein substrates underscores the protease involvement in numerous physiological roles such as cytoskeletal remodeling, particularly documented in the case of cell movement (for review see [26,27]), cell cycle progression [28], and apoptosis [29]. Moreover, the overactivation of proteases is associated with pathophysiological processes, in both acute and chronic conditions, including cardiac or neuronal ischemia, spinal cord injury, type-2 diabetes, muscular dystrophies, cardiovascular diseases [15,30], and neurodegenerative diseases [31]. Interestingly, Reg-1α is involved in many of these pathologies.

In the present study, we demonstrated that Reg-1α is a new substrate of calpain-2 in vitro with a cleavage site between residues Gln^4^ and Thr^5^, upstream from the previously identified trypsin site. We have also found that the cleavage of Reg-1α by calpain-2 is hampered by the glycosylation of the protein on Thr^5^. Moreover, using structural investigations we showed that compared to trypsin, calpain-2-cleaved Reg-1α does not form fibrils. We discuss these results in terms of structural analysis of the calpain cleavage site, as well as the possible role of Reg-1α (depending on its glycosylation status)/calpain-2 tandem in the central nervous system.

## 2. Results

### 2.1. Reg-1α Is a Calpain Substrate In Vitro

To determine whether calpain-2 directly cleaves Reg-1α, we performed in vitro cleavage reactions by incubating purified calpain-2 (see Section 4, Appendix A) with a human Reg-1α (MW = 17 kDa), expressed in bacteria as a NH_2_ terminally His-tagged protein (Figure 1A). 

In agreement with the quick protease action in vivo, we found that Reg-1α was cleaved within 15 min using a ratio of 1/10 (enzyme/substrate, *w*/*w*) generating a large fragment of about 14 kDa (Figure 1B), as observed using SDS-PAGE. The test of several ratios (enzyme/substrate) showed that the cleavage was almost complete at 1/10 within a period of 15 min, and was incomplete at 1/20 and 1/50 even after 45 min. Moreover, the absence of immunological detection of the large fragment with an anti-Histidine tag antibody (Figure 1C) at 5 and 15 min demonstrated that the cleavage site was located in the NH_2_ terminal part of the Reg-1α molecule. The presence of a band in the region of 30–32 kDa, visible after immunoblotting, with an anti-Reg-1α, suggests intact (asterisk) and cleaved Reg-1α dimers (arrowhead), as previously described [13,14]. Thus, Reg-1α appeared as a new substrate of calpain-2.

Considering that the possible cleavage of Reg-1α by calpain-2 implies that the two proteins directly interact which each other, we performed chemical crosslinking experiments using the purified inactive form of calpain-2 (mutant C105S referred as C2I) (see Section 4, Appendix A), and carbodiimide associated with *N*-hydroxysuccinimide (EDC-NHS) as bifunctional reagent. The advantage of such a mutant where the active cysteine has been changed to a serine is the ability to solely test the interaction between the two proteins. Figure 2 shows the result obtained after SDS-PAGE and western blot using both Reg-1α (Figure 2A) and calpain-2 antibodies (Figure 2B). As observed, the immunodetection with both antibodies of crosslinking products above 100 kDa (compatible with the sum of the molecular weight of the two proteins, red frame/arrowheads) demonstrated that calpain-2 interacts with Reg-1α compared to Reg-1α alone (Reg-1α-Reg-1α). 

Indeed, these new entities are compulsorily the result of the interaction of the two proteins and cannot be due to the sole crosslink of calpain-2 subunits (80 + 21 kDa), product which would react only with the anti-calpain-2 antibody. We assumed that the observed covalent products corresponded to the calpain-2 catalytic subunit-Reg-1α (80 + 17), the whole calpain-2-Reg-1α (100 + 17 kDa) or calpain-2 catalytic subunit-Reg-1α dimer [80 + (2 × 17)] due to the proximity of the molecular weight of the last two entities. We also observed the presence of oligomeric forms of Reg-1α, only revealed by the anti-Reg-1α (Figure 2A, multiple asterisks), naturally present as dimer and trimer (time 0), and oligomeric forms of superior range (time 15 and 30 min) due to the crosslink experiment. Moreover, the interaction is already visible at 15 min, which is compatible with the quick action of calpain-2 on Reg-1α (Figure 1B).

### 2.2. Identification of the Calpain Cleavage Site on Reg-1α

In order to test whether the result obtained in vitro using recombinant protein from bacteria origin would exist in vivo, we first used the software package GPS-CCD (calpain cleavage detector) developed by Liu et al. [32], recently considered to outperform other predictors for calpain-specific cleavage sites [33]. As shown in Figure 3 with the human sequence of Reg-1α, we found a higher rate for a cleavage site located between Gln^4^ and Thr^5^ (Figure 3A), regardless of the presence of the His-tag (Gln^16^-Thr^17^) (Figure 3B) or the signal peptide (Gln^26^-Thr^27^) (Figure 3C).

As the prediction site is different from the known trypsin site, between Arg^11^-Ile^12^, we next verified this prediction using MALDI-TOF mass spectrometry. We submitted the recombinant protein digested by calpain-2 to Rapiflex MALDI-TOF (Matrix-Assisted-Laser-Desorption-Ionisation Time-Of-Flight) using both linear (*m*/*z* 7000–25,000) (Figure 4A) and reflectron mode (Figure 4B,C) (*m*/*z* 500–3000). We observed, after calpain-2 cleavage, that the original peak at *m*/*z* 17,886 (inset in Figure 4A) was changed to 15,820 in the linear mode (Figure 4A) with the appearance of fragments at 1983.0 and 2004.9 in the reflectron mode (Figure 4B). The analysis of the sequence showed that the cleavage of the bacterial form occurred between the Gln^16^-Thr^17^ at the origin of the peptide MKHHHHHHASHMQEAQ of predicted value 1982.87481 [M + H]^+^, which is less than 0.13 Da away from the experimental one. We observed on a larger scale (Figure 4C) additional peaks at 2004.9 [M + Na]^+^ and 2020.9 [M + K]^+^ which are visible in the reflectron mode, and correspond to sodium and potassium adducts, respectively. Thus, these data confirm the prediction that calpain-2 cleaves the peptide bond Gln^4^-Thr^5^ of the original sequence.

### 2.3. Calpain-2 Is Not Able to Cleave the Glycosylated Form of Reg-1α

As mentioned earlier, the human Reg-1α has a Thr^5^-O-linked glycan that could influence the cleavage of the protein by calpain-2. Considering the proximity of the glycosylation with the potential calpain cleavage site, we tested the cleavage of the recombinant protein from bacterial origin, deprived of any glycosylation, and compared it with a eukaryotic form of Reg-1α (expressed in HEK cells) with possible glycosylation. We first analyzed the eukaryotic form using MALDI-TOF with linear mode and found multiple peaks, indicating that the eukaryotic form contained multiple glycan derivates (Figure 5A) compared to the bacterial form (Figure 4A, inset). Having checked that the eukaryotic form was glycosylated, we performed calpain-2 and trypsin cleavage on both forms of Reg-1α and analyzed the results using SDS-PAGE (Figure 5B) and MALDI-TOF (Figure 5C,D). As shown in Figure 5B using SDS-PAGE, calpain-2 cleavage was observed only with the bacterial form (lanes 2 and 4). With this technique, it is not possible to distinguish between the full-size protein and the form deprived of only four amino acids in the case of the eukaryotic form (lanes 3 and 4). However, we could observe that this isoform displayed a slightly higher molecular weight compared with the bacterial form (lane 3 vs. lane 1). Regarding trypsin cleavage, both forms of protein appeared cleaved by the protease (lanes 6, 8 compared with lanes 5, 7). 

Thus, in order to assess if the presence of glycosylation impedes the cleavage of the eukaryotic form of Reg-1α, we compared MALDI profiles in reflectron mode before and after calpain-2 cleavage of the eukaryotic Reg-1α (Figure 5C,D). The results did not reveal any change in the profiles of the full length in the linear mode (data not shown) or the presence of any fragment at the predicted value of 475.21 [M + H]^+^, corresponding to the expected peptide QEAQ (Figure 5D). In the latter case, we could expect a cyclization of the free Gln to 5-oxoproline (pGlu), commonly observed with ionization technique [34], which predicted value would be 458.18 due to the loss of ammonia (17,026 Da). In this molecular range (MW = 400–500 Da), none of these two fragments (475.21 and 458.18) were observed. However, three small peaks, namely 444.9, 460.9, and 476.9 both observed with the non-treated (Figure 5C) and the calpain-2-treated eukaryotic Reg-1α (Figure 5D) are certainly the fact of the matrix, although we cannot exclude impurity in the original protein preparation. We also observed the presence of a peak at *m*/*z* 358.0 after calpain-2 cleavage (Figure 5D), which was identified as E-64 (predicted value of 358.21, [M + H]^+^), the irreversible inhibitor of calpain used to stop the cleavage reaction. The analysis of the trypsin cleavage of Reg-1α using MALDI-TOF showed that the pancreatic protease cleaved both bacterial (Appendix A) and eukaryotic (Appendix A) forms at the predicted position [12], i.e., between Arg^11^ and Ile^12^ regardless of the protein source. Taken together, our results show that calpain-2, contrary to trypsin, cleaves Reg-1α only when the protein is non-glycosylated.

### 2.4. Calpain2-Cleaved Reg-1α Does Not Form Fibrils

Reg-1α has been described to form fibrillar aggregates after N-terminal truncation with trypsin, showing fibrils organized as quadruple-helical filaments (QHF) [13,14,35]. The discovery of a possible calpain cleavage of Reg-1α regulated by glycosylation raises questions regarding fibril formation. The idea was thus to test the formation of fibrils with both the unglycosylated and glycosylated form of Reg-1α post calpain-2 or trypsin cleavage. 

We used transmission electron microscopy experiments (TEM) to test the fibril formation with both recombinant Reg-1α proteins (bacterial and eukaryotic) in conjunction with the effect of calpain-2 treatment, compared to trypsin. As shown in Figure 6A, using both isoforms of Reg-1α, we did not observe any fibril formation 1 h after the calpain-2 treatment (middle panels, left), compared to the same experiment performed using trypsin (middle panel, right). As expected from previous studies with trypsin [12,13], the fibrils were of various lengths, from tens of nanometers to several micrometers. They were clearly visible after 1 h, and appeared organized in large bundles of 100–200 nm width after long incubation time (Figure 6A, right panel, 8–10 days) with Reg-1α of bacterial (top panel) compared to eukaryotic origin (bottom panel), indicating a possible influence of glycosylation on supramolecular organization of the fibrils. On the other side, even after a long period of calpain-2 incubation and regardless of the origin of Reg-1α, we did not observe fibril formation (data not shown).

Finally, we were interested in testing the formation of fibrils with the truncated form after the elimination of the cleaved NH_2_ sequence. Indeed, the fibrils previously found were observed after trypsin cleavage, in the presence of the undecapeptide [12] which could influence fibril formation. Thus, taking advantage of the presence of an His-Tag in the N-terminal part of Reg-1α of bacterial origin, we purified the truncated molecules after calpain-2 (Reg-1α^ΔN1−4^) or trypsin (Reg-1α^ΔN1−11^) cleavage, and verified their purity using both SDS-PAGE (Figure 6B) and MALDI analysis (Appendix A). In a similar manner, electron microscopy using these truncated molecules showed the presence of fibrils only in the case of Reg-1α^ΔN1−11^ obtained after trypsin cleavage (Figure 6C, bottom compared to top panels). Of note is the tendency of the fibrils to pack when Reg-1α^ΔN1−11^ was 8–10 days old (Figure 6C bottom right panel). Taken together these results demonstrate that the calpain2-treated Reg-1α does not form fibrils as observed after trypsin cleavage. 

## 3. Discussion

In the present study, we identified calpain-2 as the potential protease of Reg-1α in vitro and revealed a unique cleavage site for the protein in its NH_2_ terminal, between Gln^4^ and Thr^5^, which is distinct from the previously reported one for trypsin [36]. This type of proteolysis, typical of calpains, called modulatory proteolysis, consists in regulating substrate functions by limited proteolysis. Doing so, it is considered as one of the most essential posttranslational modifications (PTM) of proteins. Our result is strengthened by the demonstration of the direct interaction between the substrate and the protease. Moreover, the result of such cleavage being regulated by the O-glycosylation of the protein shows that both PTMs are likely to play a central role in Reg-1α function. Finally, the difference in fibril formation observed between the calpain- and trypsin-cleaved Reg-1α raises questions in the context of neurodegenerative diseases where calpains are overexpressed.

The identification of the calpain cleavage site in Reg-1α benefited from the in silico prediction tool GPS-CCD [32], based on sequence alignment, and was confirmed by mass spectrometry using the most advanced MALDI-TOF system. Several reports have pointed out that the cleavage of calpain substrate would be regulated by both primary and high-order structures (disordered regions), which explains the limited action of calpain [37,38]. Indeed, looking at calpains, the active-site cleft was found to be deeper and narrower than that of papain. Due to this constraint, it is assumed that the substrate must be in a fully extended conformation with its backbone stretched. This finding explains calpain preference for proteolyzing inter-domain unstructured regions [30]. From this point of view, Reg-1α with its 13 first residues disordered [39] appeared as a good/potential candidate for calpain cleavage.

Considering the Reg-1α cleavage site, significant amino acid preferences were found to extend over 11 residues around the scissile bond (noted P4 to P’7, see Table 1) where Pro was shown to dominate the region flanking the cleavage site [38]. The authors also showed that the segment C-terminal to the cleavage site resembles the inhibitory region of calpastatin, the specific endogenous inhibitor of calpains, acting as a substrate analog. We have thus analyzed (Table 1) the NH_2_ terminal sequence of human Reg-1α and found that 3 amino acids in the P’ position (P’1, P’4, and P’7) were conserved with calpastatin. The Gln^4^ and Glu^6^ found in the P1 and P’2 positions correspond to the preferred amino acid position for calpain cleavage [33,37,38]. The presence of a hydrophobic residue in the P2 position, namely Ala in the Reg-1α sequence, is frequently observed [40]. Moreover, considering the high frequency of cleavage by calpains around position 11 from the NH_2_ terminal [37], one could speculate that the presence of an His-tag next to the calpain cleavage site could influence the natural cleavage of Reg-1α. The prediction using the GPS-CCD ruled out this possibility showing that regardless of the presence of a His-tag, the cleavage site between Gln^4^ and Thr^5^ showed the higher score. Altogether, this analysis corroborates our data.

As part of the cleavage site, the influence of O-glycosylation, naturally occurring on the unique Thr^5^ of Reg-1α, was investigated. The demonstration that the eukaryotic protein, shown to be glycosylated, would not be cleaved by calpain-2 underscored the importance of such PTM as a means of regulation. O-glycosylation has long been reported to confer resistance to proteolysis [41], although a recent study using in silico analysis pointed out that it could be more complex than initially reported [42]. Moreover, it is generally accepted that the closer the O-glycan resides to the scissile bond, the greater the impact is on cleavage [43]. A well-known example of such regulation is brought by the prevention of an ectodomain shedding of membrane proteins due to O-glycosylation [44,45]. Reg-1α belongs to the 12% of glycosylated proteins which are exclusively O-glycosylated [46]. It is noteworthy that, although no distinct consensus sequence is known, O-glycosylations were shown to be favored in Pro-rich sequences, with a particular importance of the presence of a Pro at +3 [47]. Note that this Pro corresponds to P’4 in the preferred amino acids for calpain cleavage (Table 1). 

Our results clearly question the impact of Reg-1α cleavage by calpain-2 depending on its glycosylation status, the formation of Reg-1α fibrils, and the impact of the undecapeptide in such structures. Trypsin was already shown to cleave Reg-1α at the origin of fibrils resembling QHF [13], without demonstration of the influence of glycosylation of Reg-1α on such cleavage. In our study, we showed for the first time that Reg-1α can be cleaved by trypsin regardless of the presence of glycosylation, demonstrating that such PTM does not influence this specific cleavage or fibril formation. We also noticed that the presence of the undecapeptide seemed to influence the packing, since no large bundles were observed when the experiment was performed using purified Reg-1α^ΔN1−11^ molecules (Figure 6A,C). Moreover, we showed that the fibrils formed post-trypsin cleavage after an extended period (8–10 days), with the glycosylated Reg-1α displaying a slightly different organization (with no large packing) when compared with the large bundles observed within the same period using the non-glycosylated form of Reg-1α (Figure 6A, top and bottom right panels). The presence of glycosylation on the undecapeptide could inhibit such packing, compared to the non-glycosylated peptide, possibly by steric hindrance. This result suggests that glycosylation could affect not only the calpain cleavage of Reg-1α, but also the supramolecular organization of fibrils. Altogether, we can speculate that both glycosylation and the presence of the undecapeptide generated post-trypsin cleavage could influence the 3D organization of the fibrils.

As trypsin is not present in the brain, the identification of calpain-2 as a novel protease for the cleavage of Reg-1α could thus be of great interest. Considering our previous results on the role of Reg-1α in the central nervous system, the study of the impact of Reg-1α cleavage by calpain-2 appears pertinent for a better understanding of the structure–function relationship of Reg-1α in both physiological and pathological contexts. Indeed, we previously showed that during development the whole molecule was able to promote neurite outgrowth, and that this was the fact of the undecapeptide action [6,48]. In situations where the amount of glycosylation is altered and calpains overactivated, such as neurodegenerative diseases or diabetes, calpain cleavage of Reg-1α would generate two new entities: i.e., a peptide of only four amino acids, and the Reg-1α^ΔN1−4^ fragment. As we demonstrated, such fragments—being deprived of any capacity of fibril formation—could have different functions compared to their trypsin homologs. Moreover, Reg-1α could be the target of calpain-2 before glycosylation occurs since the calcium protease has been located in the ER [49], and the signal peptide does not hamper its action as predicted using GPS-CDD. This increased cleavage would lead to intracellular accumulation of the Reg-1α^ΔN1−4^. Further investigations will be required to test the intracellular effect of the calpain-2 cleavage of Reg-1α depending on its glycosylation status, on model of neurons overexpressing Reg-1α, glycosylated or not, and in conditions where calpains will be overactivated.

In conclusion, we have shown that Reg-1α is a new substrate of calpain-2 depending on its glycosylation status, and that this cleavage, in opposition to trypsin, does not allow fibril formation in vitro. The decrease in glycosylation processes associated with the hyperactivation of calpain-2, and the overexpression of Reg-1α observed in Alzheimer disease, prompt us to further investigate the calpain-2/Reg-1α crosstalk as well as the role of this new cleavage and its regulation by glycosylation in the context of neurodegenerative diseases.

## 4. Materials and Methods

### 4.1. Materials

Bacterial His-tagged Reg-1α (Bact. Reg-1α) was purchased from Biovendor, and the eukaryotic form (Euk. Reg-1α) with no tag was from Bio-Techne (R&D systems, Noyal Chatillon sur Seiche, France). E-64 was purchased from Sigma-Aldrich (St Louis, MO, USA) and Ni-NTA agarose from Qiagen (Les Ulis, France).

### 4.2. Calpain-2 Purification (80/21kDa)

Recombinant active rat Calpain-2 and its inactive form (mutant C105S) were expressed in *E. coli* using the co-expression of the two subunit cDNAs (pET24d and pACpET24 with double selection kanamycin + ampicillin respectively, a generous gift from Pr P.L Davies), corresponding to the whole catalytic subunit (80 kDa) plus the regulatory subunit deprived of the N-terminal glycine-rich region (21 kDa), which is equivalent to the natural autolysis product. The co-expression procedure has been shown to be necessary for the proteinase activity. The heterodimer was purified as described [50,51] with minor modifications. In our procedure, the bacteria were lysed using a mechanical breaker (1500 bars/2 passages). The purified active and inactive proteins are presented in Appendix A.

### 4.3. Gel Electrophoresis and Immunoblotting 

SDS-PAGE was performed using a Mini-PROTEAN Tetra Cell apparatus (Bio-rad) on 13.5% (or specified under Figure legends). Coloration was obtained using either colloidal Coomassie Blue (InstantBlue, Expedeon, Abcam, Cambridge, UK) or the Stain-Free™ system including addition of 2,2,2-trichloroethanol (Sigma, T54801, 1:100) in the polyacrylamide gel. After migration, gels were activated by UV exposition (45 s) and pictured with Chemidoc system (Bio-Rad, Marnes-La-Coquette, France). For western blot analysis, SDS-PAGE gels were transferred (1 h at 100 V) to PVDF membranes (0.22 µm; Bio-Rad) by standard electroblotting. Membranes were blocked for 1 h with 3% BSA in PBS containing 0.05% (*v*/*v*) Tween 20 followed by 1 h incubation with primary antibodies at room temperature. Primary antibodies were rabbit anti-Reg-1α (Abcam, Cambridge, UK, ab47099, 1:1000), anti-Calpain-2 (Cell Signaling, Ozyme, Saint Quentin Yvelines, France, 2539, 1:1000), and mouse anti-His (Sigma, Saint Quentin Fallavier, France, H1029, 1:3000). After 5 washes of 8 min with PBS/Tween, membranes were incubated for 1 h with peroxidase-conjugated goat anti-rabbit (Sigma, Saint Quentin Fallavier, France, A6154, 1:2000) or anti-mouse IgGs (Sigma, A4416, 1:2000); finally, after 5 washes, immunolabeling was revealed by chemiluminescence reaction using the ECL western blot detection reagents associated with Chemidoc System (Bio-Rad). 

### 4.4. Mass Spectrometry

Molecular masses of the proteins/peptides were determined by MALDI-TOF (matrix-assisted laser desorption/ionization time-of-flight) mass spectroscopy on a Rapiflex (Bruker Daltonics, Bremen, Germany) mass spectrometer in positive linear mode and reflectron mode. Mass spectra were acquired in the mass ranges of *m*/*z* 500–3000 and 5000–25,000 for the reflectron and linear modes, respectively. Intensity and number of laser shots were adjusted to obtain an optimal signal-to-noise ratio; most data resulted from 5000 shots at a laser frequency of 5000 (linear)–10,000 (reflectron) Hz. External calibration was performed with Bruker peptide and protein standard kits. The samples were mixed 1:1 (*v*/*v*) with a solution of sinapinic acid (SA) at 10 mg/mL or a saturated solution of alpha-cyano-4-hydroxycinnamic acid (HCCA) for linear and reflectron modes, respectively. Peptide sequences were identified by manual analysis of fragment ions and subsequent comparison of predicted (ProtParam tool—Expasy for linear mode and sum of the monoisotopic mass of the amino-acids for the reflectron mode) and experimentally-obtained fragment patterns. 

### 4.5. Cross-Linking Experiments

Reg-1α and calpain-2 in its inactive form (mutant C105S, Reg-1α-C2I, molar ratio of 2/1) or Reg-1α alone (Reg-1α-Reg-1α) were incubated in buffer 50 mM Hepes, pH 7.3, 100 mM NaCl, 1 mM DTT, 1 mM EGTA (buffer A). The complexes were cross-linked with 10 mM EDC/NHS. The reactions were allowed to proceed for 30 min, and reactions terminated by the addition of an SDS-PAGE loading buffer and heating at 95 °C during 4 min. The cross-linked species were separated by gel electrophoresis (4–15%) and further analyzed by immunodetection using both anti-Reg-1α and anti-calpain-2 antibodies.

### 4.6. Protein Cleavage 

Non glycosylated form (bact. Reg-1α) of Reg-1α, diluted in 50 mM Hepes, pH 7.3, 1 mM DTT, 1 mM EGTA was treated with calpain-2 using a ratio of enzyme/substrate varying from 1:50 to 1:10 (*w*/*w*) for a maximum of 45 min. Calpain-2 activation was obtained after addition of 4 mM CaCl_2_. The reaction was stopped by the addition of SDS-PAGE loading buffer and heating at 95 °C during 4 min. For samples used for MALDI analysis, the cleavage was performed using a ratio of 1:10 (*w*/*w*) during 30 min and the reaction was stopped by the addition of 3 µM E-64. For trypsin cleavage, the protein diluted in 50 mM Tris-HCl, pH 8 was treated with trypsin (Promega) at a ratio of 1:100 (*w*/*w*), during 30 min. For samples used for TEM, the calpain-2 or trypsin cleavages were performed using a ratio of 1:10 (*w*/*w*) or (1:100), respectively, during 1 h or 8–10 days (8–10 D).

### 4.7. Preparation of the Reg-1α^ΔN^ Molecules

After cleavage of the bacterial form of Reg-1α with an His-tag in the NH_2_ terminus either by calpain-2 or trypsin (as described elsewhere), the mixtures were incubated 1 h at room temperature with Ni-NTA agarose beads (previously washed with PBS/Tween) using a gentle agitation. Finally, after centrifugation 30 s at 10,000× *g*, the pellet containing the His-tag associated to the cleaved peptide was discarded, and the supernatants containing the Reg-1α^ΔN^ molecules were collected and a fraction of each preparation was loaded on polyacrylamide gels.

### 4.8. Transmission Electron Microscopy

A drop (8–10 µL) of 20 µM solution of the various Reg-1α samples was applied to a Formvar/carbon-coated copper grid (Agar Scientific, Gometz-La-Ville, France), negatively stained with freshly filtered 2% uranyl acetate, dried and viewed using a JEOL JEM-1400Flash (JEOL Europe SAS, Croissy Sur Seine, France) electron microscope operating at 80 kV.

## Figures and Tables

**Figure 1 ijms-23-08591-f001:**
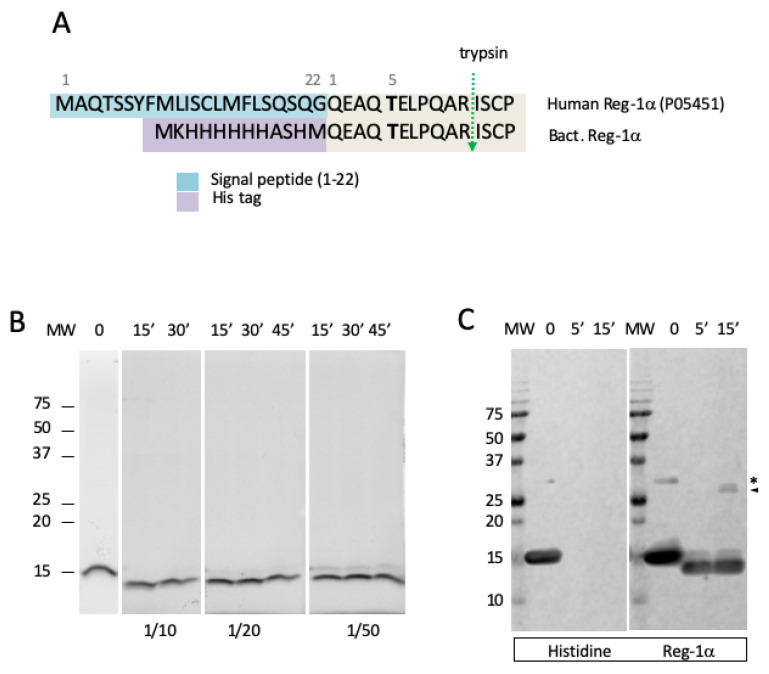
Reg-1α is a new calpain-2 substrate in vitro. (**A**) NH2 terminal sequence of human Reg-1α, including the signal peptide (blue box) compared to the isoform expressed in bacteria including the His-tag (purple box). The glycosylated Thr is noted in bold, and the position of the trypsin cleavage is indicated by a green arrow. (**B**) Intact and cleaved molecules with samples corresponding to the test of various ratio (*w/w*) of enzyme/substrate (1/10, 1/20 and 1/50) during different periods of time (0, 15, 30 and 45 min) were analyzed on 13.5% acrylamide gel and visualized using the Stain-Free Imaging Technology (see Section 4). (**C**) The uncleaved/cleaved (enzyme/substrate ratio 1/10) products were examined by western blotting using anti-Histidine and anti-Reg-1α antibodies. Note the presence of dimers (*) and cleaved dimers (arrowhead) which resist complete reduction.

**Figure 2 ijms-23-08591-f002:**
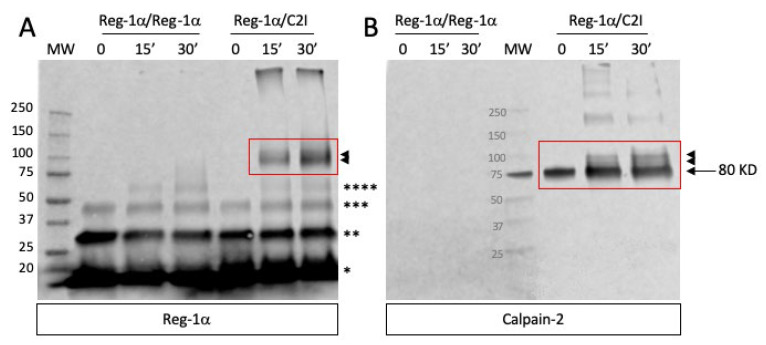
Interaction of Reg-1α with calpain-2. The cross-linking reactions (after 0, 15, and 30 min), namely Reg-1α + calpain-2 in its inactive form (Reg-1α /C2I) (red frame) and Reg-1α alone (Reg-1α /Reg-1α) were carried out as described in Section 4. SDS-PAGE was performed on a 4-15% acrylamide gel, and the presence of cross-linked products was evidenced after transfer of the proteins on a PVDF membrane by double immunodetection with anti-Reg-1α (**A**) and anti-calpain-2 (**B**) antibodies. The covalent products are indicated by arrowheads. Note the presence of Reg1α monomers (*), dimers (**), trimers (***) initially present and tetramers (****) visible only after the crosslinking reaction of Reg-1α alone.

**Figure 3 ijms-23-08591-f003:**
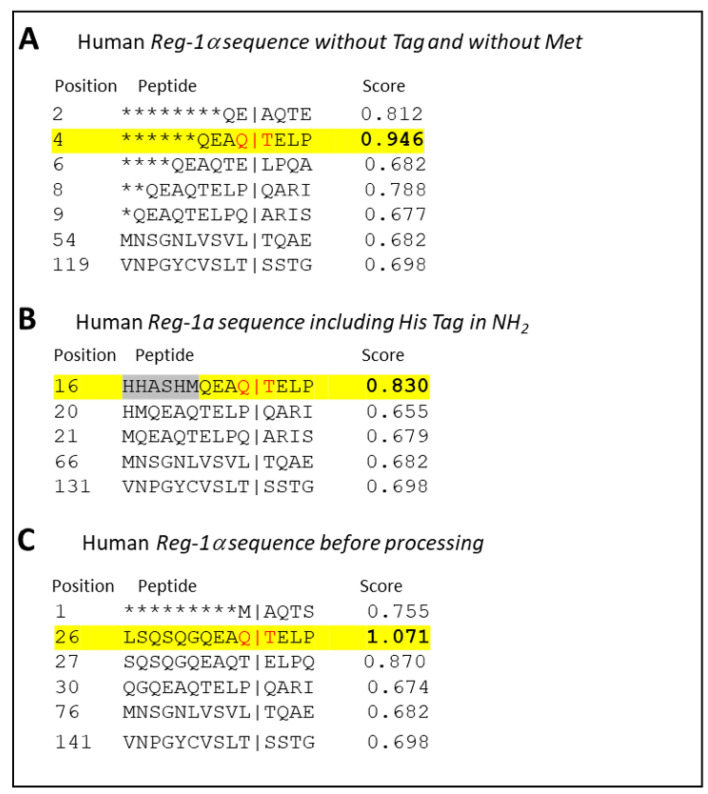
Prediction of the calpain-2 cleavage site of Reg-1α using GPS-CCD (calpain cleavage detector). (**A**,**B**) Result obtained using the GPS-CCD software [32] with the sequence of the recombinant human Reg-1α without (**A**) or with (**B**) the NH_2_-terminal His-tag and initial Met (natural protein/recombinant protein expressed in bacteria) and (**C**) with the human Reg-1α sequence, including the signal peptide (before processing). The position of the calpain-2 cleavage site predicted with the higher score is highlighted in yellow and the exact position noted in red. Note that the results of the score were obtained with the higher threshold.

**Figure 4 ijms-23-08591-f004:**
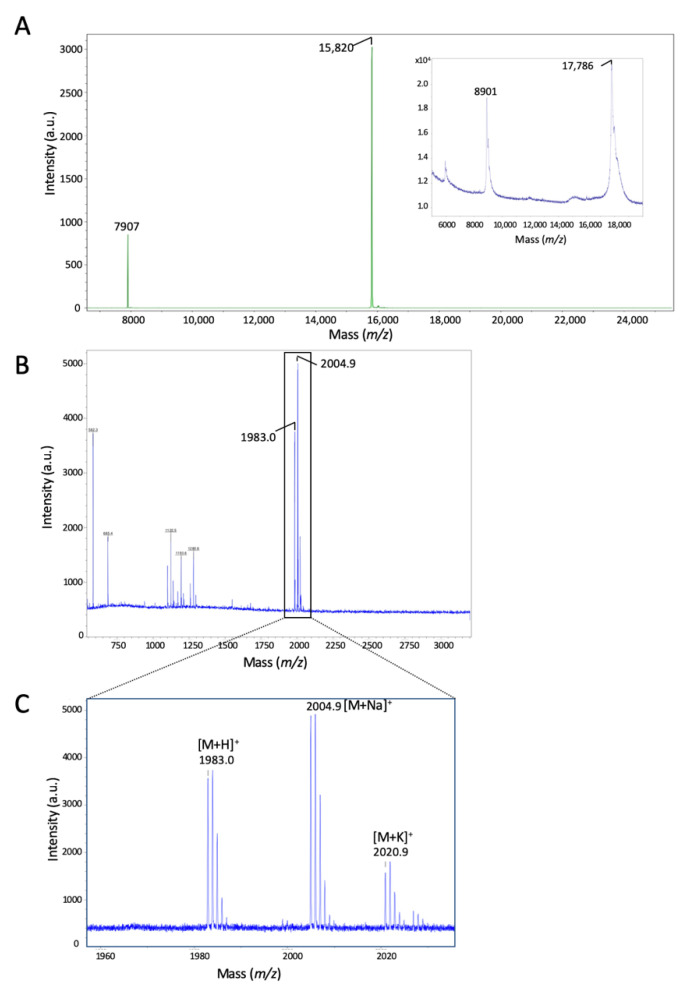
Identification of the calpain-2 cleavage site using MALDI-TOF mass spectrometry. (**A**) MALDI-TOF spectra showing Reg-1α treated with calpain-2 in the linear mode (*m*/*z* values from 7000 to 25,000). Inset, profile of the uncleaved protein and (**B**) the cleaved fragment in the reflectron mode (*m*/*z* 600–3000). (**C**) Enlargement of the spectrum of (**B**) obtained in the region of *m*/*z* 1960–2030. Note the presence of both sodium [M + Na]^+^ and potassium [M + K]^+^ adducts. The molecular weights, in Da, are shown above the peaks of each major entity.

**Figure 5 ijms-23-08591-f005:**
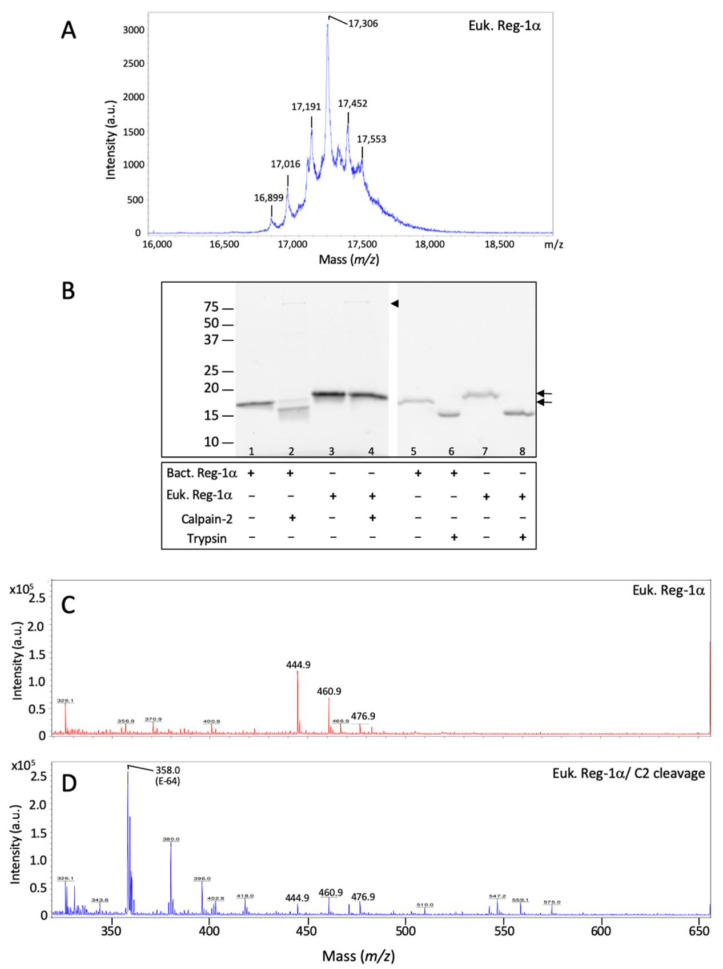
O-glycosylation of Reg-1α protects it from the calpain-2-mediated cleavage in vitro. (**A**) MALDI spectra of the eukaryotic Reg-1α (Euk. Reg-1α) obtained in the linear mode (*m*/*z* 16,000–19,000); (**B**) Both recombinant isoforms obtained from eukaryotic (Euk. Reg-1α) and bacterial (Bact. Reg-1α) sources were tested for calpain-2 (lanes 1–4) or trypsin cleavage (lanes 5–8) (ratio 1:10 and 1:100), respectively; samples were analyzed on 13.5% acrylamide gel and visualized using the Stain-Free Imaging Technology (see Section 4). Note that the protein from eukaryotic source displays a slightly higher molecular weight (see arrows) compared to the protein from bacteria due to the presence of glycosylation; arrowhead indicates the presence of calpain-2 (lanes 2,4). (**C**,**D**), MALDI reflectron mode spectra (*m*/*z* 300–650) of the glycosylated form of Reg-1α (Euk. Reg-1α) before (**C**) and after calpain-2 cleavage (Euk. Reg-1α /C2) (**D**). The position of the calpain inhibitor (E-64) used to stop the reaction is indicated.

**Figure 6 ijms-23-08591-f006:**
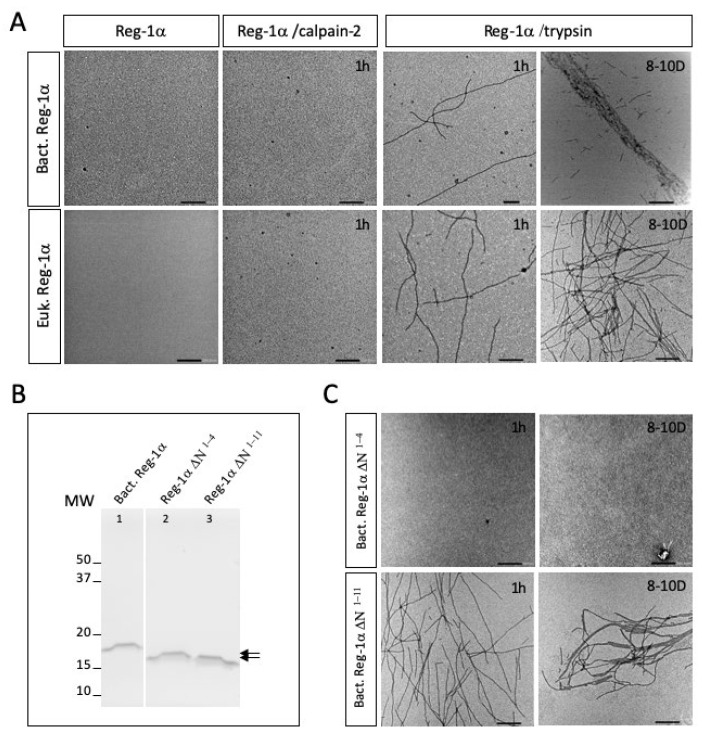
Reg-1α fibrils are only formed post cleavage of the undecapeptide. (**A**) The two recombinant forms (Bact. or Euk. Reg-1α) of the protein were analyzed after calpain-2 or trypsin cleavage at 1h and 8–10 days (8–10D) using electron microscopy. (**B**) Gel electrophoresis of the COOH-terminal domains of Reg-1α (Reg-1α∆N molecules) after purification using Ni-NTA-agarose beads (see Section 4). *Lane 1,* intact Reg-1α from bacteria source; *lanes 2–3,* supernatant obtained post Ni-NTA resin incubation after calpain-2 (Reg-1α∆N1-4) or trypsin (Reg-1αΔN1-11) cleavage. Note the difference in the molecular weight of the two purified fragments (double arrow). (**C**) Electron micrographs of the purified Reg-1α∆N molecules, post calpain-2 (top panel) and trypsin (bottom panel) cleavage after 1 h and 8–10D. Scale bars: 200 nm.

**Table 1 ijms-23-08591-t001:** Sequence alignment of Reg-1α NH_2_ terminus around the cleavage site with a sequential preference matrix of calpain according to [38].

	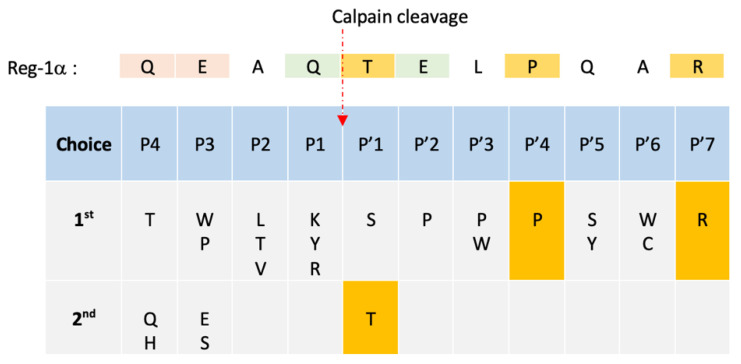
	Amino acid corresponding to the consensus Calpastatin inhibitory segment in positions P’1–P’7
	2nd choice of amino acid preference
	Xth choice of amino acid preference according to [33,37,38]

P1–P4 represent the preferred amino acid positions located upstream the scissile bond whereas P’1–P’7 represent the ones located downstream the scissile bond. Xth represents, according to [38], the 6th and 9th choice for P’2 and P1 positions, respectively; 2nd choice for P’2 according to [33]; lastly, according to [37], 3rd and 2nd choice for P’2 and P1, respectively.

## Data Availability

Not applicable.

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
