# Peer review of "Reg-1α, a New Substrate of Calpain-2 Depending on Its Glycosylation Status"

_ijms, 2022, doi:10.3390/ijms23158591_

Round 1

Reviewer 1 Report

In the manuscript "Reg-1α, a New Substrate of Calpain-2 Depending on its Glycosylation Status", Lebart et al. show with in vitro experiments that non-glycosylated Reg-1α is a new calpain-2 substrate.

Although the effects described are interesting, there are still some important points that need to be addressed by the authors:

Major:

11.)    The biggest issue is that the authors did not reconcile the different cellular localisations of Calpain-2 and Reg-1α. While Calpain-2 is expressed intracellularly, Reg-1α is secreted. How can both proteins come in contact with each other in vivo under physiological or pathophysiological conditions? Without this explanation the fact that Calpain-2 can cleave Reg-1α has no relevance.

22.)    Are there natural occurring mutations in Reg-1α known, which may affect the O-glycosylation and cleavage by Calpain-2 ?

Minor:

1)      1) The authors see putative dimers of Reg-1α on an SDS gel. No dimers should be seen in a reducing SDS-PAGE. But maybe the SDS and reducing agent is not sufficient? Please describe the buffer composition in detail. If there is not enough reducing agent, reduced cysteines may re-oxidise over time. This could lead to non-specific intermolecular disulphide bond formation by chance, which has nothing to do with physiological dimer formation.

Author Response

We are grateful to the referee for his suggestions and comments. You will first find enclosed the answer to the major points:

11) concerning the cellular localisations of the two proteins, the referee is right, calpain-2 is mainly intracellular (although an extracellular location has also been reported, mainly by Letavernier et al.) and Reg1-a is secreted. But Reg1-a also shows an intracellular localisation as shown in our previous studies (Acquatella-Tran Van Ba et al., 2012; Moussaed et al., 2018). Indeed, Reg-1 is mainly localized in the soma and along neurites in PC12 and Neuro2A cells and in primary hippocampal neurons. Additionally, in order to be secreted, the protein follows the ER-golgi pathway. During this process, calpain and Reg1-a could be found in the same cellular compartment. In particular, calpain-2 has been located in the ER (Hood et al., 2004). One could thus hypothesize that Reg-1a could be the target of the calcium protease before glycosylation occurs, especially considering that the signal peptide does not hamper the cleavage as predicted using GPS-CDD. In situation of calpain overactivation, this increased cleavage would lead to intracellular accumulation of the Reg-1aΔN1-4.

If the referee think this is an important information, we can add part of the answer at the end of the discussion (as shown below in yellow), with addition of one citation.

“As trypsin is not present in the brain, the identification of calpain-2 as a novel protease for the cleavage of Reg-1a could thus be of great interest. Considering our previous results on Reg-1a role in the central nervous system, the study of the impact of Reg-1a cleavage by calpain-2 appears pertinent for a better understanding of the structure-function relationship of Reg-1a in both physiological and pathological contexts. Indeed, we previously showed that during development the whole molecule was able to promote neurite outgrowth and that this was the fact of the undecapeptide action [6,48]. In situations where the amount of glycosylation is altered and calpains overactivated such as neurodegenerative diseases or diabetes, calpain cleavage of Reg-1a would generate two new entities, ie a peptide of only four amino acids and the Reg-1aΔN1-4 fragment. As we demonstrated, such fragments - being deprived of any capacity of fibril formation - could have different functions compared to their trypsin homologous. Moreover, Reg-1acould be the target of calpain-2 before glycosylation occurs since the calcium protease has been located in the ER (Hood et al., 2004) and the signal peptide does not hamper its action as predicted using GPS-CDD. This increased cleavage would lead to intracellular accumulation of the Reg-1aΔN1-4.Further investigations will be required to test the intracellular effect of the calpain-2 cleavage of Reg-1a depending on its glycosylation status on model of neurons overexpressing Reg-1a, glycosylated or not and in conditions where calpains will be overactivated.”

22) concerning natural occurring mutations of Reg-1a:

The natural occurring mutations in Reg-1a are essentially located in the promoter region according to Mahurkar et al. (2007). For those affecting the glycosylation, looking on UNIPROT concerning REG1A-human (P05451) and the list of existing mutations, only one mutation was reported on T27 (mutated in R) which is equivalent to Thr5 (after removal of the signal peptide) with the indication of “NO DISEASE ASSOCIATION”. We can suppose that such mutation would affect the cleavage due to its immediate proximity.

Moreover, in pathological conditions (such as diabete, Alzheimer disease) (Haukedal and Freude, 2021), glycosylation rate can be affected; this would in turn influence the cleavage of Reg-1a by calpain-2.

Finally, the literature offers examples of glycosylation influencing calpain cleavage. In particular, Levine et al. (2017) reported that alpha-synuclein which has been been site-specifically O-GlcNAcylated is notably resistant to cleavage by calpain.

Minor :

1)The putative dimers on SDS-PAGE:

We have already observed such dimers, even using denaturing conditions, and this was observed with Reg-1a in previous studies (Gregoire et al., 2001; Acquatella-Tran Van Ba et al., 2012). Additionally, it is important to note that structural studies have also demonstrated the formation of Reg-1a dimers, and tetramers (Gregoire et al., 2001; Laurine et al., 2003). Moreover, I also observed dimer formation with other protein like dysdrophin in the past (Lebart et al., 1995), so I do not think it comes from the composition of the Laemmli buffer. Composition (Buffer 2x): Tris-HCl : 0.5M pH 6.8, SDS 10%, sucrose 40%, ß-mercaptoethanol 700 mM and bromophenol blue.

Concerning the table filled by the referee N1, 3 points appeared as: “must be improved”:

It concerns:

1/the research design: chemical crosslinking experiments with a cross-linker which is consider to be a “zero length” seems fully appropriate to test the interaction between the two proteins. The action of the protease was studied in vitroand we used mass spectrometry to identify the exact position of the cleavage. Finally, the electron microscopy technique that we used was commonly used to demonstrate the presence of fibrils under different experimental conditions (Driss el Moustaine et al., 2011).

2/ the methods: in order to improve the methods, some details were added to the text; in particular, concerning the Gel electrophoresis and immunoblotting: SDS-PAGE gel were transfered (1h at 100V), and concerning the Preparations of the Reg-1aDN molecules : the mention of gentle agitation when adding the beads and the supernatants containing the Reg-1a DN molecules were collected, and a fraction of each preparation was loaded on polyacrylamide gels.

3/The conclusions appear supported by the results: the two proteins interact with each other as demonstrated by chemical crosslinking experiments. Moreover, the cleavage which is possible with the bacterial form of Reg (ie with no glycosylation) appears totally impeded when we used an eukaryotic form of the protein, having demonstrated using mass spectrometry that it is glycosylated. Finally, the results obtained using electron microscopy clearly show the impact of enzymatic cleavage (trypsin and calpain) on the formation of Reg-1a fibrils (fibrils versus no fibrils, respectively) as previously described in the literature in the case of trypsin (Cerini et al, 1999).

Reviewer 2 Report

The authors found a new substrate of Calpain-2 based on immunoblotting of cross-linking reactions and mass spectrometry. The designed experiments are very clear and straight forward to get unambiguous results. In addition, the role of glycosylation in fibrillization is identified. It is common to find out interactions between proteins. In this work, the authors can precisely determine the position/residue of the cleavage and glycosylation. Overall, this work is well performed. Therefore, I recommend this manuscript to be published.

Author Response

 We are very grateful to the referee 2 for his (her) supportive comments.

Round 2

Reviewer 1 Report

Lebart et al. show in their manuscript Reg1-1a as a new substrate for Calpain-2 based on different methods. Overall, this work is publishable.